# Overview of the Mechanical Properties of Tungsten/Steel Brazed Joints for the DEMO Fusion Reactor

**Diana Bachurina \*, Vladimir Vorkel, Alexey Suchkov, Julia Gurova, Alexander Ivannikov, Milena Penyaz, Ivan Fedotov**  **, Oleg Sevryukov and Boris Kalin** 

Department of Materials Science, National Research Nuclear University MEPhI, Moscow, Kashirskoe shosse 31, 115409 Moscow, Russia; vvorkel@mail.ru (V.V.); ansuchkov@mephi.ru (A.S.); violinarus@inbox.ru (J.G.); aaivannikov@mephi.ru (A.I.); mapenyaz@mephi.ru (M.P.); fed_ivan@mail.ru (I.F.); sevr54@mail.ru (O.S.); bakalin@mephi.ru (B.K.)
\* Correspondence: dmbachurina@mephi.ru

**Abstract:** A Demonstration (DEMO) thermonuclear reactor is the next step after the International Thermonuclear Experimental Reactor (ITER). Designs for a DEMO divertor and the First Wall require the joining of tungsten to steel; this is a difficult task, however, because of the metals' physical properties and necessary operating conditions. Brazing is a prospective technology that could be used to solve this problem. This work examines a state-of-the-art solution to the problem of joining tungsten to steel by brazing, in order to summarize best practices, identify shortcomings, and clarify mechanical property requirements. Here, we outline the ways in which brazing technology can be developed to join tungsten to steel for use in a DEMO application.

**Keywords:** DEMO; tungsten; reduced activation steel; brazing; divertor; First Wall; joining



## 1. Introduction

The creation of a Demonstration (DEMO) thermonuclear reactor is the next step after the International Thermonuclear Experimental Reactor (ITER) with the final objective of a fusion reactor power plant. As a result of the increased loads as compared to ITER, many ITER solutions cannot be used in a DEMO, especially for the divertor and the First Wall (FW). This is due to higher heat and neutron fluxes. Therefore, tungsten is considered suitable as a plasma-facing material (PFM) for both the divertor and the FW. Meanwhile, steel is considered suitable as a structural material for a helium-cooled divertor [1–4] and some FW concepts [5–7]. However, the direct connection of tungsten to steel is difficult due to the difference in their coefficients of thermal expansion (CTE). Therefore, the development of a reliable joining technology is of great interest.

At the moment, diffusion bonding and brazing are the two main methodologies being considered to achieve such a joint. The first is carried out by means of hot isostatic pressing [8,9] and spark plasma sintering [10,11]. The second is carried out by means of vacuum brazing, using foils [12,13], powders [14], and liquid-forming interlayers [15]. Some of the advantages of considering brazing as a prospective technology include:

- No deformation of the materials being joined, while during diffusion bonding the steel creeps up to 8% [16];
- Full integrity of the seam if the brazing alloy and brazing mode are chosen correctly, which is important for good heat transportation;
- The possibility of simultaneous heat treatment of the base materials and the brazing procedure;
- The possible replacement of defective elements after manufacturing and during production [17];

- Low cost, i.e., no expensive equipment is required to carry out the process. There is a wide range of brazing methods available, from vacuum brazing to ohmic or induction brazing, even in air under flux, or blowing with an inert gas.

Therefore, this technology is used in the assembly of ITER components and is currently actively investigated with a view to its use in DEMO production. Meanwhile, a compensating interlayer is usually used [12,18,19] to reduce the residual stresses arising from differences in CTE. In this case, the development of brazing alloys suitable for two seams (tungsten/interlayer and interlayer/steel) is required.

This work examines the state-of-the-art solution to the problem of joining tungsten to steel by brazing in order to summarize best practices, identify shortcomings, and clarify mechanical property requirements:

- Section 2 is a review of the working conditions of a tungsten/steel joint, including heat loads (Section 2.1), understanding the mechanical property requirements using Finite Element Analysis (FEM) (Section 2.2), and other requirements aside from the mechanical properties (Section 2.3);
- Section 3 gives an overview of the recent progress made in tungsten/steel brazing (attention is mainly paid to mechanical properties);
- Section 4 presents conclusions based on the overview of Sections 2 and 3. Finally, recommendations to all authors working on tungsten/steel brazing are given.

## 2. Working Conditions and Requirements

### 2.1. Heat Loads

The most challenging application in which tungsten/steel joints are used is a helium-cooled divertor. The operating conditions of such a divertor are more extreme than the operating conditions of the FW, i.e., a heat flux of at least 10 MW/m$^2$ [20]. This concept requires a tungsten/steel joint capable of withstanding a $\approx$700 °C operating temperature [2,21,22]. As a result of the very extreme operating conditions, Eurofusion's target design was changed [23]. As a result, an ITER-type CuCrZr pipe concept was chosen for Research & Development (R&D) activities for divertor applications. However, a lot of studies have already been published on helium-cooled divertor concepts [13,15,24–27].

Tungsten surfaces are subjected to cyclic power loads by particles and radiation from plasma. It is expected that a heat load during steady-state operation will be no higher than 0.5–1 MW/m$^2$ [28,29], but a load of 7 MW/m$^2$ is expected in the inner and outer baffle regions [29]. The DEMO reactor will work in a quasi-stationary pulse mode, whereby the reactor starts up and shuts down for several cycles. The entire pulse [30] will take 7200 s, in which the plateau takes 7000 s. The number of pulses will reach 1000–10,000 cycles, which will require nearly 20 years of functionality at operating temperature. This means that not only base materials, but brazed joints must be resistant to long-term aging and creep. Additionally, the damage of plasma-facing components, due to the huge heat loads during disruptions, is of great concern in a DEMO. The power load of uncontrolled-edge localized modes (ELMs) is expected to be ~0.2–0.5 MJ/m$^2$ over 0.6 ms [31], so the tungsten temperature can change from $\approx$280 to $\approx$630 °C in 1 s several times. Thus, resistance to thermocycling is very important for any tungsten/steel joints.

### 2.2. FEM Calculations

The first and foremost requirement for any joint is to be strong enough for a chosen application. However, there are still no clear requirements for the mechanical properties of a tungsten/steel joint. Therefore, we decided to make an approximate calculation of a direct joint under minimal heat load (0.5 MW/m$^2$), using the example of the Water Cooled Lithium Lead (WCLL) FW.

We calculated a simplified part of the FW component by means of Finite Element Analysis in ANSYS Workbench 18.2. The test component and the boundary conditions for modeling are shown in Figure 1. The temperatures used are mean values, designed for cooling and breeding in the WCLL blanket. The geometry was designed based on the

parameters presented in [32]. The 2-mm-thick tungsten and reduced activation ferritic-martensitic steel (RAFM) Rusfer were modeled, and the properties of the materials were taken from [26].

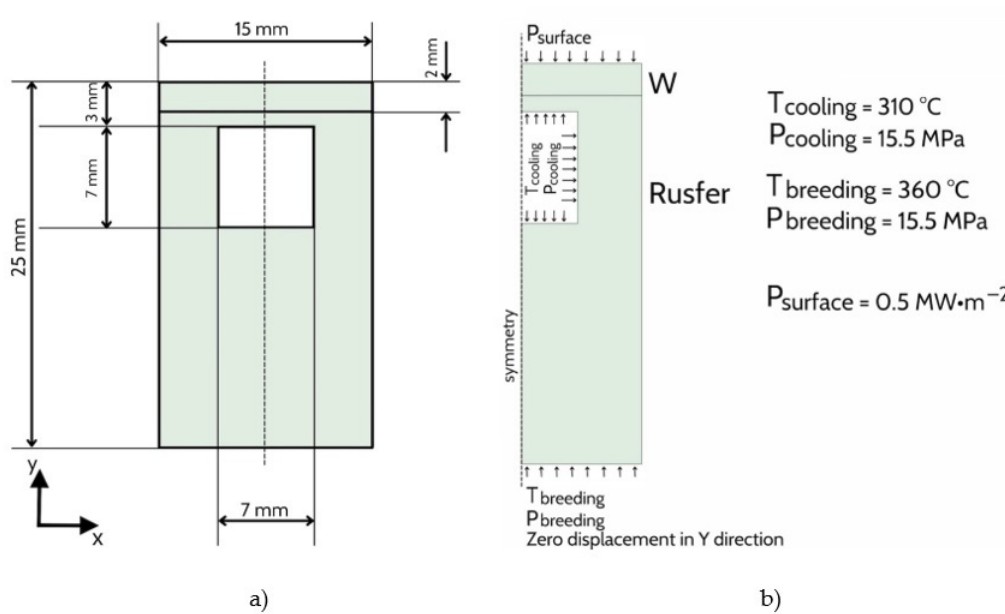

a)                                                                                                b)

**Figure 1.** Test component (**a**) and boundary conditions for FEM model (**b**).

The following load steps were modeled:

1. Cooling, from a joining temperature of 1100 °C to room temperature within 100 s;
2. Increasing the temperature and pressure of the breeder and coolant within 1 h;
3. Power ignition—with a heat load of 0.5 MW/m² on the tungsten surface within 100 s.

The mesh near the contact region for FEM calculations is shown in Figure 2. There are 10 inflation layers on each side; the smallest elements are 0.02 mm by 0.2 mm, and the growth rate in the vertical direction is 1.1.

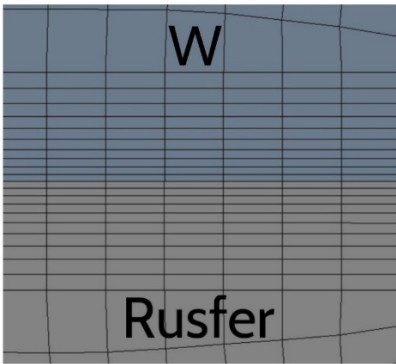

**Figure 2.** The mesh near the contact region.

The results of the FEM calculations are presented in Figure 3. Stresses $\sigma_{yy}$ were chosen to make an approximation of a required tensile strength of a tungsten/steel joint, and stresses $\sigma_{xy}$ were chosen to approximate the required shear strength. According to the FEM calculations carried out in this work, tensile strength should be no less than 224 MPa and shear strength should be no less than 180 MPa.

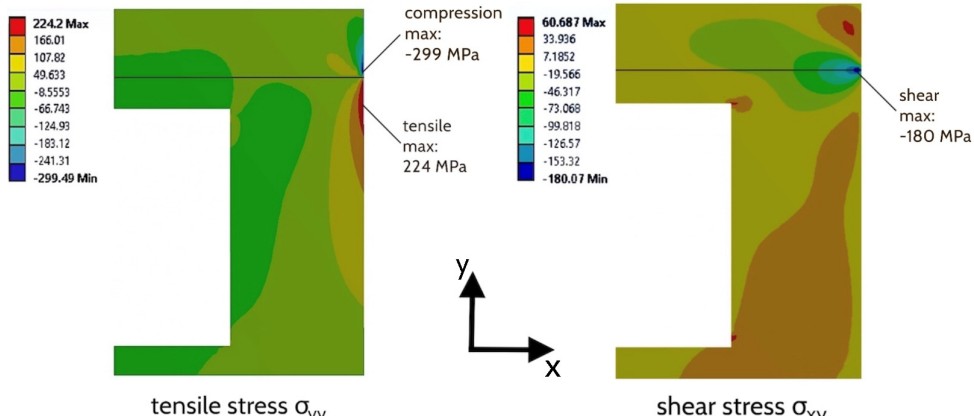

**Figure 3.** Contour plots of the stress component distributions at full heat load. Only the right half of the symmetric test component is depicted.

## 2.3. Additional Requirements

Apart from requirements for good mechanical properties, brazed joints have to be resistant to radiation damage and hydrogen retention. Furthermore, brazing alloys should meet all the requirements stated for DEMO materials. First of all, this involves reduced activation, which narrows the amount of permitted chemical elements. A specific Periodic table of the elements [33], which is shown in Figure 4, was designed based on the data given in [34]. Chemical elements that are green-colored have a residual activity of less than 10 mSv/h at 100 years from the end of operation; yellow elements have residual activity close to 10 mSv/h; and red elements have more. Hence, the latter are forbidden in a DEMO.

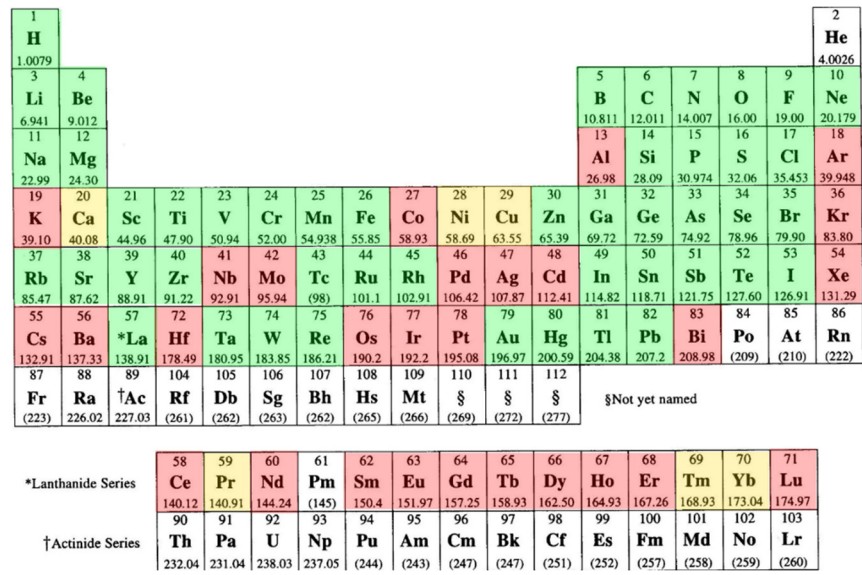

**Figure 4.** Periodic table of the elements with residual activity of the elements 100 years from the end of operation: green—less than 10 mSv/h; yellow—near 10 mSv/h; red—more than 10 mSv/h [33].

Additionally, in the brazing process, the properties of the materials to be joined should be considered. In this sense, the brazing temperature is limited, on the one hand, by tungsten's recrystallization temperature (1300 °C) [35] and, on the other hand, by the softening temperature of steel. Because the main prospective steel for DEMO applications is reduced activation ferritic martensitic steel, usually its softening temperature lies close to 550 °C, even after aging for 1 h [36]. Thus, when choosing a brazing alloy and a brazing mode, this fact should be taken into account.

The first option is to carry out brazing at a temperature well below 550 °C, though such joints will have a low unbrazing temperature. The second option is to carry out the brazing and heat treatment of steel simultaneously: either at the homogenization temperature (980–1100 °C for 30–60 min depending on the steel grade) or during aging (700–750 °C for 90–180 min).

### 3. The Latest Progress Made in Tungsten/Steel Brazing

Hereafter, we present recent updates as regards tungsten/steel joints with a focus on mechanical properties. Table 1 presents all the results obtained by brazing in relation to strength and resistance to different types of thermal loads.

Tungsten of 99.95% purity and ferritic grade steel (Fe-17Cr-0.1C, wt%) were brazed using Ta and Cu interlayers [18]. Brazing was carried out in a vacuum using amorphous foil (Ni-7Cr-5Si-3B, wt.%) with a thickness of 20 μm at 1050 °C for 1 h. Both joints were successfully achieved. Tensile tests were carried out at room temperature. The W/Ta/steel joint endured 240–275 MPa, while the W/Cu/steel joint endured 255–295 MPa.

Ma Y. et al. [13] brazed a Fe-17Cr wt.% steel/vanadium/tungsten joint with the use of a Ni-7Cr-5Si-3B-3Fe wt.% brazing alloy, at 1150 °C for 30 min. Tensile tests were carried out at room temperature. The sample withstood 143 MPa. Failure occurred in the zone with vanadium boride and Ni-V intermetallic compound.

Cai Q. et al. [27] carried out contact-reactive brazing of steel Fe-17Cr-0.1C wt.% and tungsten of 99.95 wt.% purity. A brazing composition was designed with 20 μm-thick Ti foils and 40 μm-thick Ni foil. The composition (W/Ti/Ni/Ti/steel) was brazed in a vacuum at a temperature of 1050 °C for 60 min. The paper presents the results of the microhardness measurements and tensile tests: at room temperature, approximately 185 MPa, T = 500 °C —approximately 175 MPa, T = 650 °C—approximately 150 MPa. All samples failed between nickel and steel.

Electro-chemical coatings to make liquid-forming interlayers are often used. For example, in [4], nickel and copper coatings were used. Plating nickel on tungsten provided adhesion, because copper does not interact with tungsten, and copper compensates for the mismatch in CTE. Ni was electrochemically deposited on tungsten and steel with a thickness of ≈20 μm. The filler metal Cu was deposited with a thickness of roughly 100 μm. The coated components were assembled and joined at 1100 °C for 10 min in a vacuum. The shear strength of the joint was investigated in [24] and had a value of 100 MPa.

As discussed above, Ni-based brazing compositions are often used, but its applicability according to reduced activation requirement is of concern (see Figure 1). Furthermore, it is known that radiation embrittlement of Ni alloys occurs at high temperatures (>300 °C), and transmutation into helium has a huge impact on its properties [37].

Another widespread solution is copper-based brazing compositions.

Our scientific group investigated Cu-based brazing alloys that rapidly solidified into foil. Cu-28/50Ti, Cu-12/20Sn, and Cu-12/25Ge, wt.% [12,26,33,38], and a vanadium interlayer were used to overcome the mismatch in CTE between tungsten and the Rusfer steel. In all cases, Cu-50Ti brazing alloy was used to join W with V, because Cu, Sn, and Ge do not dissolve in W, while Ti has a good solubility in W. Hence, to join V to Rusfer, other brazing alloys were investigated. The highest shear strength was achieved using Cu-28Ti brazing alloy—205 ± 12 MPa (1) obtained at 1100 °C for 20 min in brazing mode; 173 ± 25 MPa (2) obtained at 1100 °C for 60 min, plus aging at 720 °C for 180 min in brazing mode. The second mode was used to carry out simultaneous brazing and heat treatment of the Rusfer steel. Thermocycling, between 700 °C and water quenching, was applied 50 times. The shear strength of the joint dropped to 126 ± 40 MPa (1) and 50 ± 8 MPa (2).

Prado et al. have been studying the use of Cu-20Ti wt.% powder brazing alloy for joining tungsten to Eurofer steel directly [14,39–43]. The filler was fabricated by laminating a mixture of pure metallic powders or mechanically alloyed powders with an organic binder. By means of pure powders at a brazing mode of 960 °C for 10 min, the shear strength of the joint was measured as 145 ± 4 MPa [39]. When mechanically alloyed

powders are used, the value is significantly lower—93 ± 28 MPa [41]. At the same time, the microstructures of the seams are very close, but the difference in the amount and shape of the phases formed may be the reason for such a difference. The authors also evaluated the effect of annealing, which is a stage of heat treatment of Eurofer—780 °C for 90 min [14]. It turned out that the strength decreased to 60 MPa after annealing. High Heat Flux Tests were carried out in [40]. It was shown that heating up to 600 °C led to failure even after 79 cycles. The shear strength after heating up to 400 °C and 1000 cycles appeared to be 135 MPa, and after heating to 500 °C and 1000 cycles, it was 75 MPa.

Liu W. et al. [19] carried out contact-reactive brazing: W/Ti/Cu/Ti/steel Fe–17Cr–0.1C wt.% steel under hot isostatic pressing at 100 MPa. The hot isostatic pressing experiment was carried out at 1050 °C for 60 min. This joint demonstrated a very high shear strength of 248 MPa.

Peng L. et al. [44] carried out brazing at 900 °C using Cu-22TiH$_2$ filler to join pure tungsten to SS301 austenitic steel. The filler was prepared by ball milling mixtures of copper and TiH$_2$ powders. The joint demonstrated the shear strength of 98 ± 21 MPa. This result is close to that achieved by Prado et al. and no positive effect associated with H was discussed by the authors.

As it was shown in [45], ductile materials like Cu and Ni reduce residual stresses more efficiently than materials with a medium CTE (V, Ta). Pure copper as a brazing alloy was first presented in [33], though copper alloys had been widely investigated before. When the 0.1-mm-thick copper was used as a brazing alloy, in brazing conditions of 1100 °C for 20 min, the shear strength was 206 MPa and 102 MPa after thermocycling between 720 °C, with water quenching. Prado et al. [46], meanwhile, showed that the use of a 0.25-mm-thick pure Cu interlayer and 1135 °C brazing temperature gives 309 ± 32 MPa strength, but such samples were characterized by only ≈170 MPa after recovery tempering at 760 °C for 90 min. It should be noted, however, that applying temperatures of 1135 °C to the RAFM steel could lead to grain growth, which cannot be recovered by tempering at 760 °C for 90 min. This is the highest value of the shear strength of a tungsten/steel joint. Though copper does not interact with tungsten, microstructural investigations showed that steel components interact with tungsten and form a phase based on the W-Fe compound.

**Table 1.** The latest progress made in tungsten/steel brazing ($\sigma_u$—ultimate strength, $\tau$—shear strength, "-"—no data, RT—Room Temperature).

| Ref. | Brazed Joint, wt.% | Brazing Mode, °C/min | Mechanical Properties As-Received, MPa | Thermo-Mechanical Properties, MPa |
|---|---|---|---|---|
| [18] | W/brazing foil Ni–7Cr–5Si–3B/Ta 0.5 mm/brazing foil Ni–7Cr–5Si–3B/steel Fe–17Cr–0.1C W/brazing foil Ni–7Cr–5Si–3B/Cu 0.5 mm/brazing foil Ni–7Cr–5Si–3B/steel Fe–17Cr–0.1C | 1050/60 | $\sigma_u$ = 240–275<br><br>$\sigma_u$ = 255–295 | -<br><br>- |
| [13] | W/brazing foil Ni–7Cr–5Si–3B–3Fe/V 0.3 mm/brazing foil Ni–7Cr–5Si–3B–3Fe/steel Fe–17Cr | 1150/30 | $\sigma_u$ = 143 | - |
| [27] | W/Ti/Ni/Ti foils of liquid forming interlayer/steel Fe-17Cr-0.1C | 1050/60 cooling to 650/120 | $\sigma_u$ = 185 (RT) $\sigma_u$ = 175 (500 °C) $\sigma_u$ = 150 (650 °C) | - |
| [24] | W/Ni/Cu/Ni electro-chemical plated liquid forming interlayer/steel | 1100/10 | $\tau$ = 100 | - |

**Table 1.** *Cont.*

| Ref. | Brazed Joint, wt.% | Brazing Mode, °C/min | Mechanical Properties As-Received, MPa | Thermo-Mechanical Properties, MPa | | |
|---|---|---|---|---|---|---|
| [26] | W/brazing foil Cu-50Ti/Rusfer | 1100/20 | - | | | Cracks after 30 cycles |
| [33] | W/brazing foil Cu-50Ti/V 0.2 mm/brazing foil Cu-12Ge/Rusfer | 1100/20 | $\tau = 106$ | | | $\tau = 60$ |
| | W/brazing foil Cu-50Ti/V 0.2 mm/brazing foil Cu-25Ge/Rusfer | 1100/20 | $\tau = 126$ | | | $\tau = 47$ |
| [12] | W/brazing foil Cu-50Ti/V 0.2 mm/brazing foil Cu-12Sn/Rusfer | 1100/20 | $\tau = 140$ | Thermocycling: 700 °C—water quenching 50 cycles | | $\tau = 35$ |
| | W/brazing foil Cu-50Ti/V 0.2 mm/brazing foil Cu-12Sn-0.4P/Rusfer | 1100/20 | $\tau = 84$ | | | $\tau = 28$ |
| | W/brazing foil Cu-50Ti/V 0.2 mm/brazing foil Cu-20Sn/Rusfer | 1100/20 1100/60 cooling to RT + 720/180 | $\tau = 160$ $\tau = 93$ | | | $\tau = 46$ |
| [38] | W/brazing foil Cu-50Ti/V 0.2 mm/brazing foil Cu-28Ti/Rusfer | 1100/20 1100/60 cooling to RT + 720/180 | $\tau = 205$ $\tau = 173$ | | | $\tau = 126$ $\tau = 50$ |
| | W/brazing foil Cu-50Ti/V 0.2 mm/brazing foil Cu-20Ti/Rusfer | 1100/20 | $\tau = 98$ | | | $\tau = 30$ |
| [39,40] | W/Cu-20Ti pure powders with binder/Eurofer | 960/10 | $\tau = 145$ | High Heat Flux Tests 1000 cycles | 400 °C 500 °C 600 °C | $\tau = 135$ $\tau = 75$ failure after 79 cycles |
| [14,41] | W/Cu-20Ti mechanically alloyed powders with binder/Eurofer | 960/10 960/5 960/5 cooling to RT + 780/30 | $\tau = 93$ $\tau = 100$ $\tau = 60$ | - | | |
| [19] | W/Ti/Cu/Ti foils of liquid formings interlayer/steel Fe–17Cr–0.1C | 100 MPa hot isostatic pressure 1050/60 | $\tau = 248$ | - | | |
| [44] | W/Cu-22TiH2 powder paste/SS301 | 700/30 to decompose TiH2 + 900/10 | $\tau = 98$ | - | | |
| [33] | W/brazing foil of pure Cu 0.1 mm/Rusfer | 1100/20 | $\tau = 260$ | Thermocycling: 700 °C—water quenching 50 cycles | | $\tau = 102$ |

**Table 1.** *Cont.*

| Ref. | Brazed Joint, wt.% | Brazing Mode, °C/min | Mechanical Properties As-Received, MPa | Thermo-Mechanical Properties, MPa | |
|---|---|---|---|---|---|
| [46] | W/brazing foil of pure Cu 0.25 mm/Eurofer | 1135/10 1135/10 cooling to RT + 760/90 1110/10 | $\tau$ = 309 $\tau$ = 180 $\tau$ = 204 | - | |
| | W/brazing foil of pure Cu 0.05 mm/Eurofer | 1135/10 1135/10 cooling to RT + 760/90 1110/10 | $\tau$ = 225 $\tau$ = 150 $\tau$ = 193 | - | |
| [15] | W/electrodeposited Sn and Fe on Ti foil: Sn/Fe/Ti/Fe/Sn/CLF-1 | Sn thickness, mg/cm$^2$ 0 0.57 1.12 1.55 2.12 2.7 | 1090/5 cooling to 900/30 | $\tau$ = 220 - - $\tau$ = 250 - - | Thermocycling: 700 °C—water quenching 30 cycles | Failure after 2–3 cycles - - No failure No failure No failure |
| [47] | W/brazing foil Fe-3B-5Si/ODS K1 | 1200/30–240 | - | - | |
| [25] | Monocrystalline W/powder Ti-22.5Cr-7.5 V-(2–3)Be/Ta 0.1 mm/powder Fe-18Ta-(6–10)Ge-(0–4)Si-(0–2)Pd-(2–3)B/ODS Eurofer97 | 1150/20–60 | - | Thermocycling: 750 °C/20 min and air cooling/3–5 min No failure | |

One of the alternative options to copper and nickel was presented by W. Zhu et al. [15], where electroplated fully reduced activation Ti-Fe-Sn coating was used to join tungsten to CLF-1 steel. Brazing was carried out at 1090 °C for 5 min. The work details efforts to discover the necessary amount of tin. It was shown that the shear strength of the joint with 1.55 mg/cm$^2$ endured 250 MPa, but the tests were apparently carried out on only one sample, which cannot provide a reliable experimental result. Thermocycling tests were carried out according to the following regime: 30 heating cycles at 700 °C and holding for 5 min in a vacuum, followed by water quenching. It was shown that joints with amounts of tin less than 1.55 mg/cm$^2$ fail before reaching 30 cycles. The shear strength of this joint is high enough, but it is necessary to measure the temperature of unbrazing due to tin.

It is worth considering various studies on brazing oxide-dispersion-strengthened (ODS) RAFM steel, because brazing by fully reduced activation brazing alloys was carried out.

Oono N. et al. [47] carried out the brazing of tungsten to ODS K1 steel with the use of a Fe-3B-5Si wt.% brazing alloy, at 1200 °C for 30–240 min. Unfortunately, no mechanical properties were investigated.

Monocrystalline tungsten was brazed to ODS-Eurofer steel in [25] with the use of a tantalum interlayer. The brazing alloys used were powder alloys Ti-22.5Cr-7.5V-(2–3)Be wt.% for the W/Ta seam and Fe-18Ta-(6–10)Ge-(0–4)Si-(0–2)Pd-(2–3)B wt.% for the Ta/steel seam. Brazing was carried out at 1150 °C for 20–60 min. Thermocycling tests of the brazed joints were carried out at 20–750 °C for 30 cycles. No cracks or detachments were found after thermocycling.

Thus, Fe-based and Ti-based brazing alloys with added melting-point depressants (B, Be, Si) could prospectively aid in the joining of tungsten to RAFM steel.

Summarizing the review above, only a few joints can endure the stresses calculated in Section 2.3: [15,18,19,33,38,46]. The highest strength relates to the joints brazed by pure copper. However, according to the reduced activation requirement, the applicability of this metal is under discussion.

As was discussed in Section 2.1, tungsten-steel joints will work under high heat loads, but there were just a few instances of joints that displayed acceptable mechanical properties at elevated temperatures. Meanwhile mechanical properties usually change drastically with heating. Furthermore, we want to point out that sometimes the error in a strength value is very high, so it is impossible to accurately compare results.

## 4. Conclusions

The joining of tungsten to steel is of major importance in future fusion reactor applications; however, it is hard to achieve a strong connection between them. This task is made difficult by the operating conditions and the materials requirements.

Meanwhile, there are still no clear requirements for the mechanical properties of a tungsten/steel joint. Approximate calculations at a minimum heat load of the First Wall showed that the joint should have a tensile strength of no less than 224 MPa and a shear strength of no less than 180 MPa.

A number of studies have already been carried out on this topic by authors from all over the world. The vast majority were performed using a Ni- or Cu-based brazing composition. However, their residual activity 100 years from the end of operation is high enough to prohibit their application in the near future. The strongest tungsten/steel joint was achieved using pure copper. To reduce the amount of copper used, Cu-Ti brazing alloys can be used with a compensating interlayer because of its sufficiently high shear strength.

However, there is a lack of data on fully reduced activation brazing compositions. From this point of view, joints made with Ti-Sn-Fe liquid-forming interlayers are a prospective solution, as are Fe- or Ti-based brazing alloys with B, Be, and Si additions.

Furthermore, we encourage authors to take more accurate mechanical property measurements to reduce the margin of error and to consider mechanical property measurements at elevated temperatures. Additionally, we call on authors to investigate joints not only in terms of their mechanical and thermo-mechanical properties, but also their long-term aging resistance, hydrogen retention, and radiation resistance.

**Author Contributions:** Conceptualization and supervision, B.K. and O.S.; writing—original draft preparation, D.B., A.I.; writing—review and editing, A.S.; calculations, V.V.; formal analysis, I.F.; visualization, M.P., J.G.; All authors have read and agreed to the published version of the manuscript.

**Funding:** The work was funded by The Council on Grants of the President of the Russian Federation.

**Institutional Review Board Statement:** Not applicable.

**Informed Consent Statement:** Not applicable.

**Conflicts of Interest:** The authors declare no conflict of interest. The funders had no role in the design of the study; in the collection, analyses, or interpretation of data; in the writing of the manuscript, or in the decision to publish the results.

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
