# Peer review of "Overview of the Mechanical Properties of Tungsten/Steel Brazed Joints for the DEMO Fusion Reactor"

_metals, doi:10.3390/met11020209_

Round 1

Reviewer 1 Report

The authors have made a review study on mechanical properties of tungsten/steel brazed joints.

The authors are requested to improve the manuscript based on the following points:

  1. Section 2.2. More details related with the boundary conditions should be added!
  2. Dimensions should be added to component in Fig.1
  3. The flow stress versus temperature for both materials should be added to table or graph
  4. How many elements did the FEM mesh count?
  5. Does influence of FE mesh quality was tested in the model?
  6. Section 3 needs to be corrected and extended. The text should be divided into tensile strength, hardness tests or microstructure

7.The review should be extended to include literature:

https://doi.org/10.1016/j.jmatprotec.2017.03.009

Author Response

Dear Reviewer,

Thank you very much for the time you spent reviewing our article. We have corrected it according to your comments. The green text shows comments, yellow one shows additions to the manuscript or highlights within the text.

Reviewer 2 Report

Dear Authors,

I have read your manuscript with great attention and interest. In my opinion the paper is of great value as a material to compare the properties of the W-steel braze joint obtained under various brazing conditions. What is unsatisfactory is the fact that, unfortunately, not all of the described joints were tested for strength at a temperature adequate to the expected operating conditions of the title joints. I believe that this aspect should be broadly commented on by the authors. I recommend it to publish after considering of the mentioned problem and introduction of the following changes.

line 35 "No deformation of the materials to be joined" I suggest to change on - Low deformation rate of the ...

line 37 "Full integrity of a seam ..." - On the large surface of brazed joint is a significant risk of porosity imperfection 

Line 40 "Convenient replacement" I suggest to change on possible replacement

Line 42 Brazing of W in air even under inert gas blowing is high risk operation

Regards,

Author Response

Dear Reviewer,

Thank you very much for your review, You have given very reasonable remarks.

In this file You will find replies. The green text shows comments, yellow one shows additions to the manuscript or highlights within the text.

Reviewer 3 Report

It will be good if you insert a phrase at the end of introduction indicating how this work is structured

“DEMO”, “R&D” define all acronyms

Please cite this “However, a lot of works 60 have already been published in a sense of helium cooled divertor concept.”

I have noted you have introduced a case study but no where was mentioned in abstract or introduction about this please reformulate accordingly …otherwise is difficult to understand its purpose

Which version “ANSYS Workbench”

Not very clear boundary condition in Fig 1

“According to the 95 calculations tensile strength” which calculations? I assume there was a theoretical calculations somewhere, if so please provide !

And these results of FEM requires further discussion

Overall the work is interesting but is very poor structured and I suggest to reshape it for better presentation

Author Response

Dear Reviewer,

Thank you for the comments! We have added required information according to your advice. In this file the green text shows comments, yellow one shows additions to the manuscript or highlights within the text

Round 2

Reviewer 3 Report

.